# Neural Architecture Search in Embedding Space

## Abstract

The neural architecture search (NAS) algorithm with reinforcement learning can be a powerful and novel framework for the automatic discovering process of neural architectures. However, its application is restricted by noncontinuous and high-dimensional search spaces, which result in difficulty in optimization. To resolve these problems, we proposed NAS in embedding space (NASES), which is a novel framework. Unlike other NAS with reinforcement learning approaches that search over a discrete and high-dimensional architecture space, this approach enables reinforcement learning to search in an embedding space by using architecture encoders and decoders. The current experiment demonstrated that the performance of the final architecture network using the NASES procedure is comparable with that of other popular NAS approaches for the image classification task on CIFAR-10. The beneficial-performance and effectiveness of NASES was impressive even when only the architecture-embedding searching and pre-training controller were applied without other NAS tricks such as parameter sharing. Specifically, considerable reduction in searches was achieved by reducing the average number of searching to $<100$ architectures to achieve a final architecture for the NASES procedure.

## 1 Introduction

Deep neural networks have enabled advances in image recognition, sequential pattern recognition, recommendation systems, and various tasks in the past decades. However, selecting a suitable neural architecture is frequently arduous because of the classical and new neural architectures emerging daily. In general, manual design of network architectures according to the cases is achievable. However, hyperparameter tuning and architecture engineering through manual selection requires considerable time. Furthermore, manually designing a neural network architecture requires substantial experience in deep learning.

Given the aforementioned reasons, neural architecture search (NAS), an automated architecture engineering, has been successful in the past years. The NAS algorithm is divided into three dimensions, namely search space, search strategy, and performance estimation strategy (Elsken et al., 2019). Outstanding results have been achieved using NAS with the reinforcement learning search strategy (Zoph & Le, 2017). Here, a recurrent network was used to generate a string to form a child network. However, such a type of network exhibits two problems: noncontinuous and high-dimensional search space. The frequent large strings of action from the recurrent network and the discrete space result in difficulty in optimization. The critical contribution of this study is the improvement of the dimension and quality of the search space that could provide a more efficient framework to solve the two problems of searching architectures.

If a vector that can represent network architecture without discrete values is determined, then the noncontinuous aforementioned disadvantages can be addressed. We proposed the NAS in embedding space (NASES) method, which involves mapping origin architecture to architecture-embedding by using an architecture encoder. The advantage of embedding space includes the lower-dimensional and continues space, it considerably alleviates the difficulty in the optimization problem of the NAS procedure with reinforcement learning. To learn and search on the embedding space, we developed a mechanism to generate architecture encoder and decoder to promote origin architecture communication with the embedding space, and the autoencoder network was used in the mechanism (Hinton

& Salakhutdinov, 2006). The architecture simulator simulates the origin architecture space, which assists the real architecture encoder learning. The decoder realizes the relationship between origin architecture and architecture-embedding, which maps the architecture-embedding to the origin architecture.

The NASES procedure was implemented in two stages. We obtained a pretraining architecture decoder and a pretraining architecture simulator in the first stage and provided the compression rate between the embedding size and testing loss. In the second stage, we used the NASES procedure for image classification on CIFAR-10 by using the network pretrained in the first stage. The results of the experiment were efficient and indicated that NASES was highly efficient and considerably reduced the number of searching architectures to <100 in <12 GPU hours. Thus, the results were comparable with that of other popular NAS methods.

## 2 RELATED WORK

### 2.1 REINFORCEMENT LEARNING WITH ACTION EMBEDDING

Reinforcement learning is a general approach that can be applied broadly to various areas. However, the large and discrete action space causes problems in function approximation. The majority of studies have focused on two approaches, one approach factorizes the action space into binary subspaces (Pazis & Parr, 2011; Dulac-Arnold et al., 2011). The other approach involves embedding discrete actions into a continuous action, determining optimal actions in the continuous space, and selecting the nearest discrete action to reduce the scaling of action sizes (Dulac-Arnold et al., 2015; Hasselt & Wiering, 2009).

### 2.2 SEARCH STRATEGY WITH REINFORCEMENT LEARNING

Our search strategy is based on reinforcement learning. Zoph & Le (2017) provided a novel NAS framework, which incorporated reinforcement learning and applied it to two agents of the child network and the controller of the recurrent network. The child network generated neural architecture that can be considered the action of the controller network. Unlike the use of policy gradient by Zoph & Le (2017), Williamsu (1992), and Baker et al. (2016) used q-learning to update the weight of the network.

### 2.3 NAS WITH CONTINUES VECTOR

Most NAS procedures use a discrete search space. Unlike other approaches, such as the learning over discrete and nondifferentiable search space, Liu et al. (2018) proposed an approach of differentiable architecture search (DARTS), which was based on the continuous relaxation of the architecture representation. On the basis of DARTS, Hundt et al. (2019) proposed sharp DARTS, which is a more general, balanced, and consistent design. The closest concept to NASES is the approach proposed by Luo et al. (2018) in which an encoder and a decoder was to map neural architectures in a continuous space on gradient-based optimization and a predictor was used to achieve embedding accuracy.

## 3 METHODS

In this section, to elucidate the NASES procedure, we followed the aforementioned three dimensions: search strategy, search space, and performance estimation strategy.

### 3.1 SEARCH STRATEGY

The search strategy is a search method for fast and accurate exploration of the space of neural architectures and involves techniques such as reinforcement learning, evolutionary algorithm, and gradient-based method. These are popular strategies in NAS.

In the NAS with reinforcement learning, which is generally designed with two components of the controller network and child network (Figure 1), the controller network is usually used to control the

child network architecture and generates string and the child network construct neural architecture by using the output of the controller in each NAS iteration. The controller network calculates the policy gradient to update the network by validating the performance of child network. Thus, severe penalty is imposed when performance is low. The controller that is constructed using a multilayer perceptron rather than the recurrent neural layer, generates such continuous value, rather than a string, in the NASES. This is discussed in the next subsection.

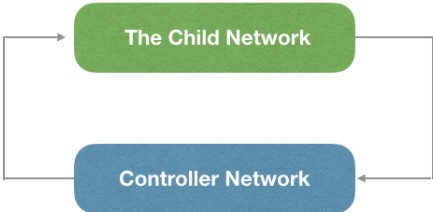

Figure 1: Overview of NAS with reinforcement learning. The child network receives information from the controller and generates a network architecture to evaluate date. The controller network receives a reward from the evaluation of the child network for updating network parameters.

## 3.2 SEARCH SPACE

The main contributions of NASES is in the search space domain, which resolves the two afore-mentioned problems of noncontinuous and high-dimensional space in reinforcement learning; these problems lead to difficult optimization. NASES is similar to the general NAS procedure, which also includes the child network and controller network. The optimization of maximize accuracy is also used as a policy gradient method. However, our method differed from the general NAS procedure; first, we developed an architecture encoder as the controller network to control the architecture of the child network and projected origin architecture into architecture-embedding. Second, we devised the architecture decoder network, which decodes architecture-embedding from the architecture controller network to the origin architecture to ensure the child network can understand and generate network by using architecture-embedding (see Figure 2). That is, to alleviate these problems, the architecture decoder functions as a translator to translate low-dimension embedding into high-dimension vector for smooth childcontroller network communication. Furthermore, the search space was bounded using micro search in this study. Thus, we did not apply the cell-based trick of hierarchical representation because we attempted to search neural network on the complete architecture and not only on the cell.

### 3.2.1 THREE PRINCIPAL FUNCTIONS OF NASES.

The NASES, has three principal functions. This section describes the functions of the architecture decoder, architecture simulator, and controller network.

**Architecture Decoder**  To obtain architecture-embedding decoder, first we created an approximate of virtual distribution transformation, which projected the low-dimension space into high-dimension space. That is, we transformed architecture-embedding into origin architecture.

$$f_\theta : \mathbb{R}^n \to \mathbb{R}^m$$
$$\boldsymbol{a} = f_\theta(\hat{\boldsymbol{a}})$$

where $f_\theta$ is a decode function parameterized by $\theta$, $R^n$ is the architecture-embedding space, $R^m$ is the origin architecture space, $\boldsymbol{a}$ is the set of origin architecture, $\hat{\boldsymbol{a}}$ is the set of architecture-embedding. This function released the controller network architecture and could be projected on another space not bounded on the origin architecture space. This function is efficient and provides distribution transformers. To develop this approximator, an architecture simulator is required, which is discussed in the next subsection.

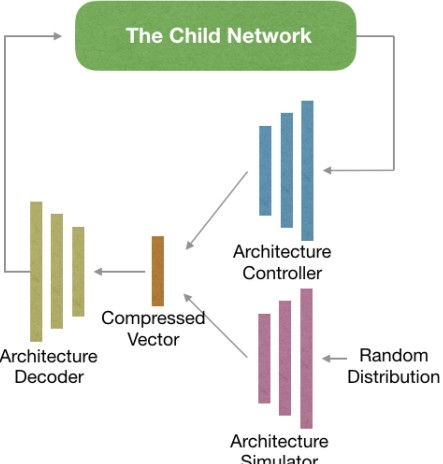

Figure 2: Overview of NAS on the embedding space. We used three functions namely architecture controller, architecture decoder, and architecture simulator, rather than the origin controller network for searching the neural architecture. Architecture controller received the origin architecture and generated architecture-embedding. Architecture decoder received the architecture-embedding and generated the origin architecture. Architecture simulator received random distribution and generated architecture-embedding.

**Architecture Simulator**    An architecture simulator is an approach of the approximator of the virtual distribution transformation; its purpose creation of a function that simulates distribution to achieve architecture-embedding. Furthermore, the distribution of simulation is not limited to discrete or continuous space.

$$s_v : U \to \mathbb{R}^n$$
$$\hat{\boldsymbol{a}} = s_v(U)$$

where $s_v$ is a function parametrized by $v$ and U is a uniform distribution.

**Obtaining the architecture encoder and architecture simulator**    To obtain the architecture encoder and simulator, we used the autoencoder network (Hinton & Salakhutdinov, 2006). The autoencoder network is a unsupervised algorithm for distribution transformation and dimension reduction. The autoencoder generates a representation by using the reduced encoding closest to its original input. Therefore, the architecture encoder and simulator were assembled in the autoencoder network. We pretrained an autoencoder network before training the controller, in which input space and target space had the same distribution. We used uniform distribution. For convenient policy gradient learning, the activation function of the middle layer used was sigmoid; which leads the output with Bernoulli distribution.

**Controller**    A controller controls the child neural architecture. The neural architecture and hyperparameters of the controller are copied from the architecture simulator. The input is the child state of network, and the output is architecture-embedding with Bernoulli distribution.

$$g_k : \mathbb{R}^m \to \mathbb{R}^n$$

where $g_k$ is an encode function parameterized by $k$, $R^n$ is the architecture-embedding space, $R^m$ is the origin architecture space, $A$ is the set of origin architecture, and $\hat{a}$ is the set of architecture-embedding. We did not retrain the controller on the NASES procedures again and only fine-tuned weights. The controller network can explore the architecture-embedding if the weights are initialized using the pretraining simulator of the autoencoder network because the pretraining simulator

can project uniform distribution into architecture-embedding. Another advantage is fast searching because the controller network is not required to learn projecting uniform distribution into architecture-embedding again and focuses on searching architectures. Moreover, we devised a new reward function by using the accuracy of the child network in training and validation. The reward is obtained only from the accuracy of the validation set of the past NAS with reinforcement learning. In this study, the reward function is different from the general function, and it not only considers accuracy on the validation set but also estimates generalization errors (Eq. 1).

$$
\begin{aligned}
Reward =& (ACC_{val} - (ACC_{tr} - ACC_{val}))^3 \\
=& (2ACC_{val} - ACC_{tr})^3
\end{aligned}
\tag{1}
$$

### 3.2.2 CHILD MODEL

The child model receives continuous vectors from the controller and generates neural architecture by using the pretrained decoder model. Here, we described the NASES mechanism for creating a network architecture. We required four hyperparameters, namely number of filters, filter size, kernel type, and connection coefficient, in a layer. More details are in Appendix A.

### 3.3 PERFORMANCE ESTIMATION STRATEGY

To reduce the computational burden, we used two approaches of obtaining lower fidelities to estimate performance. First, the learning scheduler follows cosine annealing with $l_{max} = 0.05$, $l_{min} = 0.001$ and $T_0$ = epochs (Loshchilov & Hutter, 2017). Second, each architecture search was run for 10 epochs on the search phrase, and final architecture is run for 700 epochs. The detailed is presented in Alg 1.

---

**Algorithm 1** Neural Architecture Search in Embedding Space

**Require:** uniform distribution $U$; architecture simulator $s_v$; architecture decoder $f_\theta$; controller $g$; child network $c$ number of optimization iterations $L$; number of epochs in training phase $e_1$; number of epochs in final phase $e_2$;

1: $s_v$ and $f_\theta$ were assembled in an autoencoder network
2: Train the autoencoder network and evaluate its performance by uniform distribution $U$.
3: The controller $g$ weights initialized with architecture simulator pretrained weights $v$.
4: **for** $l=1, ..., L$ **do**
5:      Generate child network c by architecture $\boldsymbol{a}$.
6:      Train and evaluate data child network $c$ in epochs $e_1$ by training and validation set.
7:      Compute reward $r$ from train and validation performance.
8:      Train controller network $g_v$ using reward $r$.
9:      Project architecture $\boldsymbol{a}$ into architecture-embedding $\hat{\boldsymbol{a}}$ by using the controller network $g_v$.
10:     Decode architecture-embedding $\hat{\boldsymbol{a}}$ into architecture $\boldsymbol{a}$ by using the architecture decoder $f_\theta$.
11: **end for**
12: Train and evaluate on the child network with the highest reward in epoch $e_2$ by training and testing set.

---

## 4 EXPERIMENTS

We describe two stages of the experiment in this section. The first stage involves training an architecture decoder and architecture simulator network and selecting an appropriate compression ratio of architecture-embedding. The second stage involves applying the result of first stage to discover novel neural architectures for image classification on CIFAR-10 (Krizhevsky & Hinton, 2009) by using NASES.

### 4.1 First Stage: Pretraining Architecture Decoder and Simulator Network

#### 4.1.1 Dataset

In the first stage of NASES experiment, our goal was to map origin architecture to architecture-embedding. We assembled the architecture simulator and decoder in an autoencoder network and trained this autoencoder network instead of training the simulator and decoder. To mimic the origin architecture space, we sampled 300000 as a training set data from uniform distribution and sampled 100000 as the testing set data. In this case, we sampled uniform distribution at the interval [0, 30].

#### 4.1.2 Training Details

The optimization of the autoencoder network was achieved by using an Adam (Kingma & Ba, 2015) optimizer with a learning rate of 0.00001. During the training of the autoencoder network, the learning schedule is increased the batch size instead of decaying the learning rate (Smith et al., 2018) and saved weights with the lowest test loss during testing. In the network architecture, the policy gradient methods update the probability distribution of actions so that the controllers actions with high expected reward exhibit a high probability for an observed state. Therefore, the activation function is a sigmoid function used in the middle layer hidden of the autoencoder network, and it leads the distribution of hidden output to Bernoulli distribution, suitable for computing the policy gradient to update the controller network. Other details regarding the experimental procedures are as follows: Three fully connected layers with the number units of 1000, 500, and 100 were used for the simulator and decoder networks; the first layer exhibited the ReLu activation function, and the second and third layers exhibited the tanh activation function. The architecture decoder network exhibited the same neural architecture and hyperparameters setting as the architecture simulator network. The loss function used a least square error.

#### 4.1.3 Result

To understand the loss of information from the embedding space, we evaluated the compression ratios of 0.83, 0.67, 0.5, 0.33, 0.17, 0.08, and 0.02 to examine the utility of compression. We set an upper bound of testing loss as a baseline, because if the decoder is incapable of decoding on architecture-embedding, the best strategy always is the predicted average value (average is 15 in this case). The utility of compression ratio on different sizes of embedding by using the autoencoder network is presented in Table 1. The testing loss represents the loss of information after compression by the autoencoder network, and the loss of information increases with the compression ratio. Appendix B, Figure 3 illustrates the double-axis plot of the testing loss and embedding size (according to Table 1). As illustrated in Appendix B, Figure 3, a definite trade-off exists between loss and compression rate; high compression rate leads to high information loss. In this case, we suggested a range of 20-30 as the appropriate embedding size by using the double-axis plot with cross-area of testing loss curve and embedding size curve. In this range, the compression rate is more than half and information loss is considerably low. For image classification (second experiment stage) we followed the experiment results of the first stage, and the set the compression rate at 0.33. That is, the origin size of 60 was projected into the embedding size of 20.

### 4.2 Second Stage: Image Classification on CIFAR-10

#### 4.2.1 Dataset

The second stage of the experiment is a multiclass classification for assigning a class to the image object. The CIFAR-10 (Krizhevsky & Hinton, 2009) data set consists of 60000 color images of 32 32 RGB in 100 classes. Each class has 6000 images with 5000 training data and 1000 testing data. Additionally, to achieve standardization and normalization, we applied only three standard data augmentation techniques: (1) Subtracting the mean and then dividing the answer by the standard deviation, which ensures that all variables have mean zero and standard deviation 1. (2) Centrally padding on training set to 40 40 and randomly cropping images back to 32 32. (3) Randomly flipping images horizontally.

Table 1: Performance of the autoencoder network. The left block represents the compression ratio. The right block represents information loss on the training and testing set.

| Origin Size | Embedding Size | Compression ratio | Training Loss | Testing Loss |
|:---:|:---:|:---:|:---:|:---:|
| 60 | 1 | 0.02 | 71.78 | 72.58 |
| 60 | 5 | 0.08 | 59.82 | 63.37 |
| 60 | 10 | 0.17 | 53.38 | 57.52 |
| 60 | 20 | 0.33 | 40.79 | 46.15 |
| 60 | 30 | 0.5 | 30.53 | 34.77 |
| 60 | 40 | 0.67 | 19.14 | 22.66 |
| 60 | 50 | 0.83 | 7.54 | 10.76 |
| 60 | 60 | 1 | 1.74 | 2.23 |

### 4.2.2 SETTINGS

The spilled validation ratio was 0.9; we then randomly split 45000 and 5000 images for training and validation, respectively, in the neural searching procedure. Finally, we used 50000 images for training and 10000 images for testing when the NASES search procedure was complete. Each architecture search procedure was run for 10 epochs on the search phase, and the final architecture was run for 700 epochs.

The child network is described in the paragraph following the method section. The hyperparameters setting of the child network considers ENAS (Pham et al., 2018) as a reference. It was trained with Nesterov momentum, the momentum of 0.9 (Nesterov, 1983). The learning schedule followed the learning rate decay with a cosine annealing for each batch ($l_{max}$ = 0.05, $l_{min}$ = 0.001, $T_0$ = epochs) (Loshchilov & Hutter, 2017), batch size of 128, weight decay of 1e-4. We initialize $w$ with $He$ initialization(He et al., 2015) in the child network. We designed a 15-layer convolutional architecture by using 60 hyperparameters ( a layer for four hyperparameters contains: number of filters, filer size, kernel type, and connection coefficient). The NASES mechanism provided effective mechanism, and it only required applying 20 hyperparameters on embedding vector by using the 0.33 compression ratio. By following the controller network perspectives, we took the pretrained parameters of an already trained model from the first phase experiment of the pretrained architecture simulator. The controller hyperparameters setting and neural architecture followed the first stage of the experiment.

According to the reward function engineering perspectives, Eq. 1 was used as our reward function; this function not only considered accuracy of the validation set but also estimated the generalization error. We stored the reward in a reward pool in each NASES iteration. Furthermore, we normalized and updated the rewards of the reward pool at the end of each iteration. To prevent the dependency of the final architecture on initial architectures, we ran random architectures ten times to collect rewards without updating parameters of the controller network in the beginning. We sampled three architecture examples from reward pool to update the controller network in each NASES iteration. Furthermore, the epsilon-greedy approach (Watkins, 1989) occurred randomly with the probability epsilon. Therefore, we have 10 % to generate architecture randomly to out of the reward pool too.

### 4.2.3 RESULT

We ran the NASES procedure five times by using different random seeds on a single Nvidia V100 GPU, and NASES required approximately 12 hours to determine the final architecture for a NASES procedure. Furthermore, the average number of searching architectures to achieve final architecture for the NASES procedure was <100, the number of searching architectures was reduced considerably compared with past NAS approaches. To determine the performance of the architecture, we evaluated the child network by using the final architecture network on the CIFAR-10 test dataset.

Table 2: Performance and GPU computing time on macro search space of the NAS approach for class classification on CIFAR-10: First block represents the results of the final network. The second and the last blocks represent the results of the final network for more filters to each layer.

| Level | Method | GPUs | Days | Params | Error |
|---|---|---|---|---|---|
| 1 | Macro NAS with Q-Learning (Zhong et al., 2018) | 10 | 8-10 | 11.2m | 6.92 |
| 1 | SMASH (Brock et al., 2017) | 1 | 1.5 | 16.0m | 4.03 |
| 1 | NAS (Zoph & Le, 2017) | 800 | 21-28 | 7.1m | 4.47 |
| 1 | NASES | 1 | 0.5 | 8m | 4.07 |
| 2 | Net Transformation (Cai et al., 2017) | 5 | 2 | 19.7m. | 5.70 |
| 2 | ENAS (Pham et al., 2018) | 1 | 0.32 | 21.3m | 4.23 |
| 2 | NASES + more filters1. | 1 | 0.5 | 20.4m | 3.93 |
| 3 | NAS + more filters | 800 | 21-28 | 37.4m | 3.65 |
| 3 | ENAS + more filters. | 1 | 0.32 | 38.0m | 3.87 |
| 3 | NASES + more filters2 | 1 | 0.5 | 28.4m | 3.71 |

Table 2 summarizes the performance of NASES and other NAS approaches by using the macro search algorithms. This final architecture is presented in Appendix C, Figure 4.

In Table 2, the approaches have been into three levels based on the number of parameters. The first block of Table 2 presents the performances of NAS approaches; the NASES final architecture that achieves 4.07 error rate on the testing set uses only 8 million parameters, which is comparable with other NAS approaches. For comparing more approaches and models, we added a number of filters to each layer of the final architecture. The second block of Table 2 represents the performance of NAS approaches when the number of parameters was approximately 20 million, and the NASES final architecture can be improved to 3.93 error rate by adding 100 to the number of filters of each layer. Finally, to evaluate the high parameter network, we added 150 to the number of filters of each layer. Notably, the NASES final architecture that achieved 3.71 error rate only used 28.4 million parameters, which was better than approximately 38 million parameters used by the ENAS (Pham et al., 2018) and NAS (Zoph & Le, 2017). NASES required approximately half GPU day to discover the final architecture. The beneficial-performance and effectiveness of NASES was impressive even when only the architecture-embedding searching and pre-training controller were applied without other NAS tricks such as sharing parameters (Pham et al., 2018).

## 5   CONCLUSION

NAS with reinforcement learning is a powerful and novel framework for the automatic discovering process of neural architectures. Here, we designed a novel NAS framework, and this approach alleviated the two problems of noncontinuous and high-dimensional search space of NAS with reinforcement learning. We named this NAS framework NASES, in which the controller can be searched on embedding space by using the architecture decoder and architecture simulator. We achieved favorable results for image classification on CIFAR-10; the NASES exhibited efficient performance and high effectiveness when the number of searching architecture was reduced to <100 architectures. We proposed a simple method to estimate the compression ratio of architecture-embedding.

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

# A  The Child Model details

## A.1  The Operations.

The 12 operations were provided to the controller by using two hyperparameters, namely filter size and kernel type. The filter size represents the amount of neighbor information during the current layer processing, and kernel type represents the components of the neural network, including the convolution layer, depthwise-separable convolution layer, maximum pooling layer, and average pooling layer. The child network receives these continuous vectors, and we developed a rule that transforms into discrete vectors. For example, in the kernel type, the rule is that $\leq 7$ assigns depthwise-separable convolution, $>7$ or $\leq 15$ assigns convolution. For filter size, the rule is that $\leq 10$ assigns to the $3 \times 3$ filter size. According to this principle, the operations available for the child network are convolution with filter sizes $3 \times 3$, $5 \times 5$, and $7 \times 7$; depthwise-separable convolution with filter sizes $3 \times 3$, $5 \times 5$, and $7 \times 7$; maximum pooling with filter sizes $3 \times 3$, $5 \times 5$, and $7 \times 7$; and average pooling with filter sizes $3 \times 3$, $5 \times 5$, and $7 \times 7$. In addition, each depthwise-separable convolution was applied twice (Barret Zoph & Le, 2018).

## A.2  Skip Connection.

The skip connections are essential connections that occur from the early layers to the later layers through addition or straight up concatenation. The reasoning behind this skip connection is that they exhibit an uninterrupted gradient flow from the first layer to the last layer, which tackles the vanishing gradient problem. In this study, the hyperparameter of the connection coefficient assigned the early layer to connect to last multiple layers with closer connection coefficient; these layers were concatenated in the channel dimension at the end of the layer. Therefore, we created another rule, and the layers are a connection when the connection coefficient is close to three; for example, the layer one and two are a connection when their connection coefficient is between two and five.

## A.3  The Order of the Blocks in Each Layer.

Performance was affected by the order of blocks in each layer. We applied the order of ReLu-conv-batchnorm (Ioffe & Szegedy, 2015). Moreover, the kernel size of $1 \times 1$ convolution filters can be applied to change the dimensionality in the filter space. We applied the order of ReLu-conv-batchnorm to $1 \times 1$ kernel size convolution layers before the convolution layer, except for the first layer.

## A.4  Global Average Pooling.

We employed a trick into the NASES of the global average pooling (Min Lin & Yan, 2013) which is an operation that calculates the average output of each feature map in the final convolution layer for reducing the number of parameters from the full connection layer.

## B    Double-axis Plot of The Testing Loss and Embedding Size

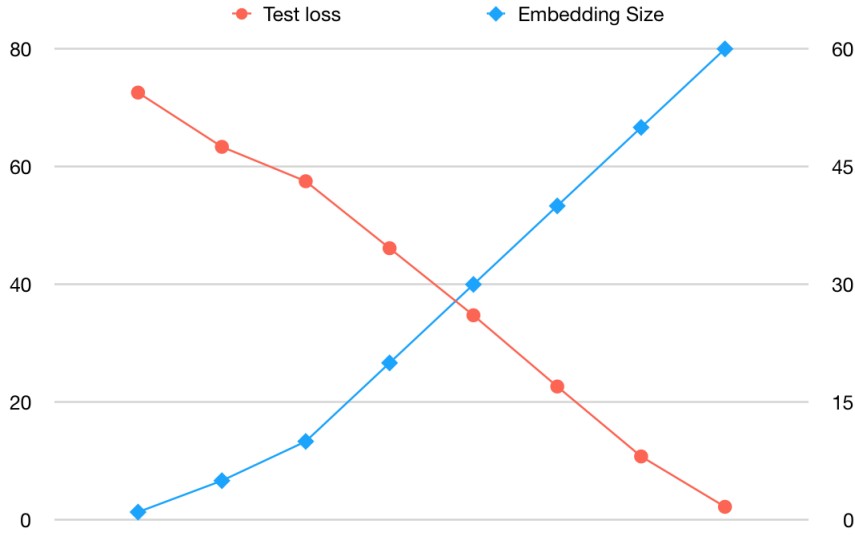

Figure 3: Double-axis diagram that illustrates the testing loss and embedding size, which offer useful information for selecting an appropriate embedding size. Right axis: testing loss. Left axis: embedding size.

## C    Final Architecture

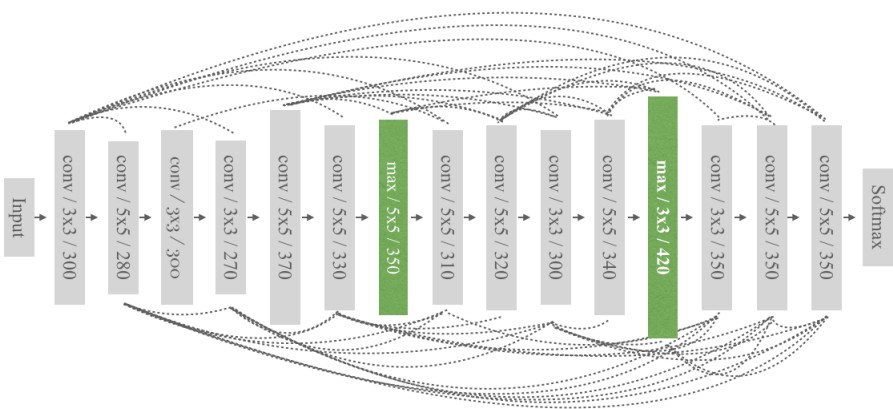

Figure 4: The NASES final architecture discovered on the macro search for image classification. It is a 15-layer convolutional architecture by using 20 NASES hyperparameters.

