# OpenReview forum: "Neural Architecture Search in Embedding Space"
_ICLR.cc/2020/Conference — Reject_

### Official Review · AnonReviewer3 · 2019-10-23
**Official Blind Review #3**

**Rating:** 1

**Review:**

This work searches neural architectures in an embedding space, which is continuous. Overall, it lacks of innovation and experimental results are not strong enough.

	1. The innovation and technical contribution are weak. The key idea of searching in embedding space has already been proposed and applied in  Luo et al. 2018. The authors do not differentiate this work from Luo et al. 2018, and I didn't find any essential differences between them.

	2. The proposed method does work well. It is not compared with latest algorithms including NAO and DARTS. The reported numbers in Table 2 are poor, far from SOTA numbers.

Besides, there are many writing issues and typos.

**Experience Assessment:**

I have published in this field for several years.

**Review Assessment: Checking Correctness Of Derivations And Theory:**

N/A

**Review Assessment: Checking Correctness Of Experiments:**

I assessed the sensibility of the experiments.

**Review Assessment: Thoroughness In Paper Reading:**

I read the paper at least twice and used my best judgement in assessing the paper.

---

> ### Author Response · Authors · 2019-11-09
> **Response to Reviewer 3**
>
> Thank for your comment,
>
> 1. This paper is very different from Luo et al. 2018. First, Luo et al. 2018 applies gradient descent in the embedded space.  We proposed another technical, which extended embedded space could be applied in NAS with reinforcement learning. Second, for the technical contribution, we droped accuracy predictor, it makes the framework more simply and avoided bias from accuracy predictor. Besides, it is very quick to train pretrained encoder and pretrained decoder by our methods.
>
>
> 2. We knew latest and excellent algorithms like DARTS and NAO (we cited both). However, we still did not compare with them, because of NASES is a method of improving search space, so we made comparisons of different technicals with macro search ( like Macro NAS with Q-Learning, Net Transformation) and other methods of improving search space (like cell-based and ENAS). Maybe the performance is far from SOTA( we didn’t apply cell-based and weights sharing and cutout), but we still can see the NASES experiments results were very impressive. For example:
>
> - It is a new method of improving search space: The experiment results were very competitive when we compare with dimension of improving search space, like micro search NAS with Q-Learning,  which spend 96 GPU days and got 3.6 error rate ( we will add into Table 2). And ENAS.
>
> - Spend up:  NAO without weights sharing(and use cell-based), it need to spend 200 GPU days. In our experiment, NASES without weights sharing( specially, it is macro search), it just only a half GPU day on more search space.
>
> - Considerable reduction in searches was achieved by reducing the average number of searching to <100 architectures to achieve a final architecture.

---

### Official Review · AnonReviewer1 · 2019-10-26
**Official Blind Review #1**

**Rating:** 3

**Review:**

Summary:
This paper borrows the idea of word-to-vector from NLP and applies it in reinforcement learning based Neural Architecture Search (NAS). It suggests a pretrained encoder to transform the search space to a dense and continuous architecture-embedding space. First it trains the architecture-embedding encoder and decoder with self-supervision learning like Auto-Encoder.  Then it performs reinforcement learning based Neural Architecture Search(NAS) in the architecture-embedding space.

Strength:
There is no architecture prior, such as cell, in the searching process. Thus it's more general and can explore more architectures possibilities.
Because it performs architecture search in a continuous space, a CNN based controller is used instead of a RNN controller.
The result of the proposed method on CIFAR-10 is comparable with other popular NAS approaches.
It reduces the number of searching architectures to <100 in <12 GPU hours without using tricks such as cell or parameter sharing.

Weakness:
The evaluations are highly insufficient. It only performs experiment on CIFAR-10, and the generalization ability on other datasets is unclear. In many NAS works. CIFAR-100 and ImageNet are commonly used to evaluate the performance.
Besides, there is no comparison with more recent and related important methods such as DARTS and the method proposed by Luo et al. (2018). Actually its performance is not as good as Darts or the best performance reported in ENAS.


**Experience Assessment:**

I have read many papers in this area.

**Review Assessment: Checking Correctness Of Derivations And Theory:**

I assessed the sensibility of the derivations and theory.

**Review Assessment: Checking Correctness Of Experiments:**

I assessed the sensibility of the experiments.

**Review Assessment: Thoroughness In Paper Reading:**

I read the paper at least twice and used my best judgement in assessing the paper.

---

> ### Author Response · Authors · 2019-11-10
> **Response to Reviewer 1**
>
> Thank for your comment,
>
> 1.  This suggestion is very useful. The reason of only performs experiment on CIFAR10 is the original plan is to only compare with different methods of improving search space (like cell-based). If we performs more dataset will be greater, but it can not deny the contribution of this paper.
>
> 2. Yes, we knew excellent algorithms DARTS and NAO (we cited both). However, we still did not compare with it, because of NASES is a method of improving search space. Besides, the performance is not as good as DARTS and ENAS due to we didn’t use cutout, weight sharing and micro search. Strive STOA performance is not ideal goal in this paper, so we only made comparisons of different technicals with macro search and other methods of improving search space (like cell-based).  For example, micro search NAS with Q-Learning,  which spend 96 GPU days and got 3.6 error rate. And macro search ENAS, which spend 0.32 GPU days and got 3.87 error rate.  AND in this paper, macro search NASES, which spend 0.5 GPU days and got 3.71 error rate ( and only use 28.4m params and number of searching to <100 architectures to achieve a final architecture).  Moreover, NAO without weights sharing(and use cell based), it need to spend 200 GPU days. Those experiment results show NASES is very competitive and excellent contribution.

---

### Official Review · AnonReviewer2 · 2019-10-28
**Official Blind Review #2**

**Rating:** 3

**Review:**

The paper proposes an interesting idea to perform Neural Architecture Search: first, an auto-encoder is pre-trained to encode/decode an neural architecture to/from a continuous low-dimensional embedding space; then the decoder is fixed but the encoder is copied as an agent controller for reinforcement learning. The controller is optimized by taking actions in the embedding space. The reward is also different from previous works which usually only considered validation accuracy but this work also considers the generalization gap.

The idea is interesting, but there are some problems on both the method's and the experimental sides:
1. NAO [1] also embeds neural architectures to a continuous space. Different from NAO which applies gradient descent in the embedded space, this paper uses RL. I double that RL can work better than gradient descent in a continuous space. The paper should compare with NAO. Ideally, this paper might work better than NAO if the accuracy predictor in the NAO is not accurate, while this paper uses real accuracy as a reward for search. However, this is not soundly compared.
2. It is unreasonable to discretize continuous actions to a Bernoulli distribution. Many RL methods, such as DDPG, can handle continuous actions;
3. The paper uses Eq. 1 as a reward. It's interesting, but it's unclear why the generalization error is needed. Ablation study is required.
4. As the community makes more progresses in AutoML, a better and better (smaller and smaller) search space is used. It doesn't make much sense to compare the search time under different search spaces. Comparison under the same setting (e.g. NASBench-101) is required.


Minors:
1. missing a number in "and T0 = epochs"
2. missing "x" in "32 32 RGB in 100 classes", and "100" should be "10"

[1] Luo, Renqian, Fei Tian, Tao Qin, Enhong Chen, and Tie-Yan Liu. "Neural architecture optimization." In Advances in neural information processing systems, pp. 7816-7827. 2018.

**Experience Assessment:**

I have read many papers in this area.

**Review Assessment: Checking Correctness Of Derivations And Theory:**

N/A

**Review Assessment: Checking Correctness Of Experiments:**

I carefully checked the experiments.

**Review Assessment: Thoroughness In Paper Reading:**

I read the paper thoroughly.

---

> ### Author Response · Authors · 2019-11-10
> **Response to Reviewer 2**
>
> Thank you for the review and for taking the time to thoroughly read and comment on this paper.
>
> 1. Yes, we knew excellent algorithms NAO(we cited it). However, we still did not compare with it, because of NASES is a method of improving search space. In this paper, strive STOA performance is not ideal goal, so we only made comparisons of different technicals with macro search and other methods of improving search space (like cell-based).  We can see the NASES experiments results were very impressive. For example:  This is very competitive when we compare with dimension of improving search space, like micro search NAS with Q-Learning,  which spend 96 GPU days and got 3.6 error rate, and ENAS. Besides, if NAO without weights sharing(and use cell based), it need to spend 200 GPU days. In our experiment, NASES without weights sharing( specially, it is macro search), it just only a half GPU day on more search space.
>
> 2.  It is our mistake we will fix it, not Bernoulli distribution, just continuous actions between 0 and 1.
>
> 3. Reward function made by generalization error due to performance estimation strategy. Our strategy is less epochs to estimate performance, so if we only estimate
> validation performance it will overfit on final architecture, because of NASES will only search for high validation performance on less epoch step( The train loss could be very low). In the final step,  there is not room for improvement when we train 700 epochs on final architecture.
>
> 4.  Yes, I agree autoML makes more progresses and better, and I agree comparison under the same setting is required, that is one of reason we didn’t compared with DARTS and NAO, we want to based on macro search and compared with different methods of improving search space. However, it can not deny contribution of NASES.  NASBench-101 is a very excellent work but parallel with our job.

---

### Decision · Program_Chairs · 2019-12-19

**Decision:**

Reject

**Comment:**

This paper proposes a method for neural architecture search in embedding space. This is an interesting idea, but its novelty is limited due to its similarity to the NAO approach. Also, the empirical evaluation is too limited; comparisons should have been performed to NAO and other contemporary NAS methods, such as DARTS.

Due the factors above, all reviewers gave rejecting scores (3,3,1). The rebuttal did not remove the main issues, resulting in the reviewers sticking to their scores. I therefore recommend rejection.